



# Multi-decadal atmospheric carbon dioxide measurements in Central Europe, Hungary

László Haszpra[1,2,3]

[1]Institute for Nuclear Research, H-4026 Debrecen, Hungary
[2]Institute of Earth Physics and Space Science, H-9400 Sopron, Hungary
[3]formerly at Hungarian Meteorological Service, H-1181 Budapest, Hungary

*Correspondence to*: László Haszpra (haszpra.laszlo@atomki.hu)

**Abstract.** The paper reviews and evaluates a 30-year-long atmospheric $CO_2$ data series measured at Hegyhátsál tall-tower greenhouse gas monitoring site, a member of WMO GAW, NOAA, and ICOS networks (id. code: HUN). The paper also
gives the technical description of the monitoring system, and that of the physical environment of the station. This low elevation (248 m above m.s.l.), mid-continental Central European site shows a $3.90\pm0.83$ µmol mol$^{-1}$ offset relative to the latitudinally representative marine boundary layer reference concentration presumably due to the European net anthropogenic emissions. The long-term trend (2.20 µmol mol$^{-1}$ year$^{-1}$) closely follows the global tendencies. In the concentration growth rate, the ENSO effect is clearly detectable with a 6-7 months lag-time. The summer diurnal
concentration amplitude is slightly decreasing due to the faster-than-average increase of the nighttime concentrations, which is related to the warming climate. The warming climate also caused a $0.96\pm0.41$ day year$^{-1}$ advance in the beginning of the summer $CO_2$-deficit season in the first half of the measurement period, which did not continue later.

## 1 Introduction

More than a century ago Ekholm (1901) and Arrhenius (1908) raised that carbon dioxide ($CO_2$) produced by coal burning
might accumulate in the atmosphere causing global climate change in the long run. Later, analyzing the sporadic measurements available, Callendar considered it proven that carbon dioxide was indeed accumulating in the atmosphere, and it might be the driver of the warming trend observed since the beginning of the 20th century (Callendar, 1938; 1949). However, the measurements were not convincing due to their technical and representativeness problems. Finally, radiocarbon studies of Suess and Revelle confirmed that atmospheric carbon dioxide concentrations were indeed increasing
(Suess, 1955; Revelle and Suess, 1957), potentially foreshadowing dangerous global climate change. Their works led to the initiation of direct continuous monitoring of atmospheric carbon dioxide concentration by Keeling in 1957 (Keeling, 1960). Since David Keeling's pioneering work, a large number of monitoring sites have been established around the world. In the beginning, they were established in very isolated remote places, far from any anthropogenic and biospheric influences (arctic regions, high mountain peaks, mid-oceanic islands). However, one of the main actors of the global carbon cycle is the
biosphere with its climate-sensitive assimilation, respiration, and carbon storage. While the remote sites were able to record





the changes in the $CO_2$ content of the global atmosphere, they could not provide detailed information on the governing biospheric processes. By the end of the 1980s, it became evident that the carbon cycle could not be properly quantified, and thus the future evolution of the climate could not be assessed, without the operation of extensive monitoring networks in the mid-continental, vegetation-covered regions (Tans, 1991).

In 1981, when the first $CO_2$ monitoring station was established in Hungary (K-puszta, 46°58'N, 19°33'E, 125 m above m.s.l. – WMO GAW code: KPS), there were only six similar monitoring sites in Europe, five in Germany (Levin et al., 1995), and one in Italy (Ciattaglia, 1983), which reported data to the World Meteorological Organization (WMO) (WDCGG, 2023a). Measurements at K-puszta started in June 1981. Tans (1991) suggested the use of tall towers for monitoring in continental environment to increase spatial representativeness avoiding the direct influence of the underlying local vegetation.

Construction of a tall tower at the original Hungarian monitoring site, K-puszta was not possible, therefore a TV/radio transmission tower 220 km to the west of it (Hegyhátsál, 46°57'N, 16°39'E, 248 m above m.s.l. –WMO GAW code: HUN) was equipped in 1993. The in situ measurements started in September 1994. By now, the two sites together have a 43-year-long data series from this Central European region. K-puszta was shut down in 1999 but allowed 5 years of parallel measurements at the two sites.

Since the 1960s, several monitoring sites have been established, especially during the last decade, and more than 150 are reporting data to international databases such as the WMO World Data Centre for Greenhouse Gases (WDCGG, 2023a) or the ObsPack product of the United States National Oceanic and Atmospheric Administration (Masarie et al., 2014; Schuldt et al., 2023) today. The global dataset is primarily used for the localization and quantification of carbon dioxide sources/sinks by inverse atmospheric transport models, as well as for the determination of the carbon budget of the 50 atmosphere. For a large number of stations, the measurement data of the individual monitoring sites are also published with the technical details of the measurements and presenting the site-specific variations in the $CO_2$ concentration (see e.g. Navascues and Rus, 1991; Cundari et al., 1995; Levin et al., 1995; Aalto et al., 2002; Derwent et al., 2002; Necki et al., 2003; Schmidt et al., 2003; Zhou et al., 2006; Artuso et al., 2009; Thompson et al., 2009; Popa et al., 2010; Winderlich et al., 2010; Vermeulen et al., 2011; Brailsford et al., 2012; Stephens et al., 2013; Fang et al., 2014; Liu et al., 2014; Schmidt et al., 55 2014; Fang et al., 2015; Lopez et al., 2015; Zhu and Yoshikawa-Inoue, 2015; Apadula et al., 2019; Conil et al., 2019; Curcoll et al., 2019; Pérez et al., 2020; Xia et al., 2020; Botía et al., 2022; Lelandais et al., 2022; Panov et al., 2022; Adcock et al., 2023; Wu et al., 2023 – and others). This information may be valuable for model tuning and validation, as well as for the interpretation of the model results. The reliability of the model results depends on the coverage, density, and spatial distribution of the monitoring network. The coverage of Europe has increased significantly during the last two decades, 60 especially with the establishment of the pan-European Integrated Carbon Observation System (ICOS) (Heiskanen et al., 2022). A large part of Europe is located in the zone of westerlies. A monitoring station in the eastern part of Europe can provide valuable information on emissions in the highly industrialized, densely populated Western Europe. Despite the development of the European monitoring network, the Hungarian monitoring sites are still the southeasternmost stations in geographical Europe.





As the characteristics of the first Hungarian monitoring site, K-puszta, have already been published (Haszpra, 1995, 1999a; 1999b), the present paper focuses on the measurements performed at Hegyhátsál tall-tower greenhouse gas monitoring station since 1994.

The instruments used at the monitoring stations provided the atmospheric dry air mole fraction of carbon dioxide. For simplicity, throughout this paper, we use the term "concentration" as a synonym.

## 2 Measurements and data

### 2.1 Monitoring site

Hegyhátsál tall-tower greenhouse gas monitoring station was established in the framework of a scientific cooperation between the Hungarian Meteorological Service (HMS) and the U.S. National Oceanic and Atmospheric Administration (NOAA) in 1993. The 117 m tall tower equipped with air intakes and meteorological sensors is a TV/radio transmitter tower

owned by the telecommunication company Antenna Hungária Corporation. The operation of the monitoring site was taken over from HMS by the Institute for Nuclear Physics (ATOMKI), Hungary, in 2020.

As shown in Fig. 1 the station is located near the western edge of the Pannonian Basin (46°57'N, 16°39'E), at an altitude of 248 m above the mean sea level, in a fairly flat region, in a rural environment, which provides a high spatial representativeness to the measurements. The terrain does not modify the large-scale atmospheric conditions, thus no special

microclimate can develop around the station. The monitoring site is surrounded by agricultural fields (dominantly corn, winter wheat, sunflower, and rape in crop rotation with small plots of other cultivars of annually changing types), grasslands, and forest patches. The distribution of vegetation types within 10 km of the tower (53 % agricultural region, 35 % forest, 6 % grassland, 6 % other [transitional woodland-scrublands, settlements, etc.]) does not differ much from the average for the surrounding Western Hungarian Landscape Unit (Barcza et al., 2009). The soil type in the region of the tower is 'Lessivated

brown forest soil' (Alfisol, according to the USDA system). The upper layer is generally 10–20 cm thick, and its organic matter content is 5–8 % (Haszpra et al., 2018).

Human habitations within 10 km of the tower are only small villages (100–400 inhabitants). The nearest village is Hegyhátsál, giving the name to the tower, located 400-1200 m to the northwest. At the time of the beginning of the measurements, the number of inhabitants was 183, which decreased to 139 by 2023 (KSH, 2023). There is no notable

industrial activity in this predominantly agricultural region. Local roads have mostly low levels of traffic (100-400 vehicles day$^{-1}$). The only major road in the region is the 2x1-lane trans-European E65 running northwest–southeast with ~4000 vehicles per day (Magyar Közút, 2019). Its closest point to the monitoring site is about 500 m to the southwest. A Lagrangian forward transport model was used to estimate the influence of the local emission sources on the concentration measurements. The study (Haszpra et al., 2022) showed that during the most critical winter periods (residential heating in the

village in addition to the road traffic, and limited vertical mixing of the atmosphere) the average excess $CO_2$ concentration at the tower is 0.0013 µmol mol$^{-1}$, and in 99.91 % of the winter periods studied the excess did not exceed 0.04 µmol mol$^{-1}$.





Another study (Haszpra et al., 2008) revealed the lack of nighttime accumulation of sulfur hexafluoride ($SF_6$) at the station. $SF_6$ is mainly used in high-voltage electrical equipment, and thus its emission through leakages is a good indicator of anthropogenic activity. The diurnal variation of the atmospheric mixing would result in a diurnal variation in the

concentration if emission were present. The lack of diurnal variation in the $SF_6$ mixing ratio indicates that there is no remarkable source within the area represented by the measurements. The station can be classified as a regionally representative monitoring site.

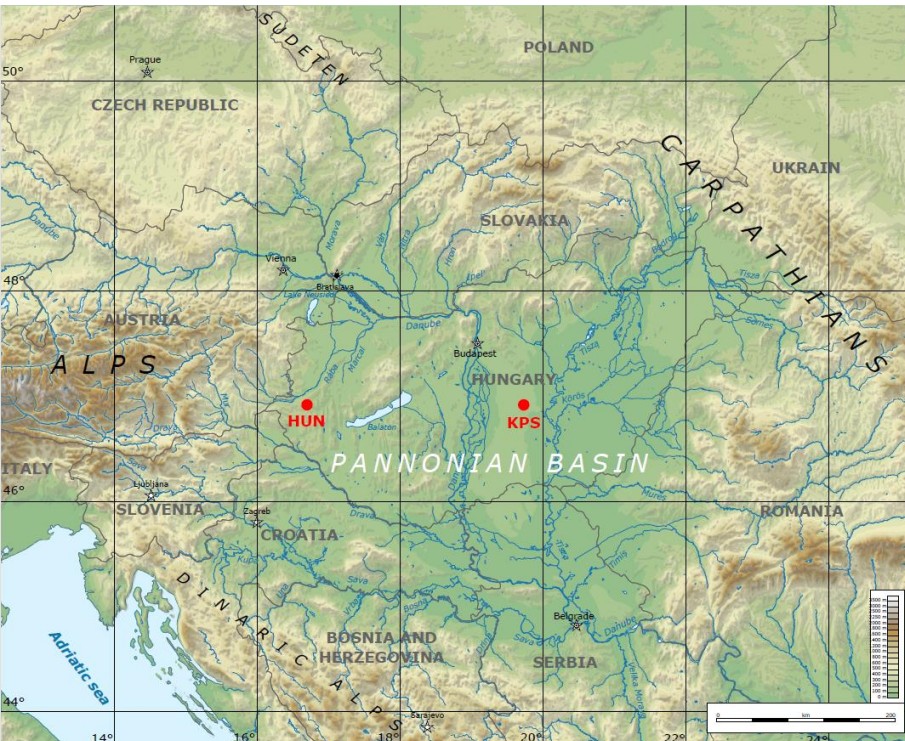

**Figure 1: Geographical locations of the Hungarian CO₂ monitoring stations (HUN – Hegyhátsál, KPS – K-puszta) (Map base: https://en.wikipedia.org/wiki/Pannonian_Basin).**

The climate of the region is a warm temperate one, fully humid, with a warm summer (Köppen-Geiger class Cfb) (WDCGG, 2023b). The mean annual temperature in the region was 10.4 °C in 1991–2020, increasing rapidly with time (0.54±0.13 °C

decade$^{-1}$). The average annual precipitation is 690 mm showing no significant trend but a wide range of variation between 475 and 939 mm year$^{-1}$ during the indicated period (Hungarian Meteorological Service, 2023) (Fig. S1). Although the monitoring station is located in the zone of westerly winds, the prevailing wind directions at the site are northeast and southwest. This is the consequence of the location of the Alps, which rise approximately 100 km to the west of the station and significantly modify the regional wind pattern. The westerly airflows bypass the Alps on the north or the south side and

reach the monitoring site almost perpendicularly. The sensitivity area map of the station calculated by the STILT model (Lin





et al., 2003) using the software tool provided by the ICOS Carbon Portal is shown in Fig. 2. The detailed characterization of the station's footprint is available on the ICOS Carbon Portal (Carbon Portal ICOS RI, 2024).

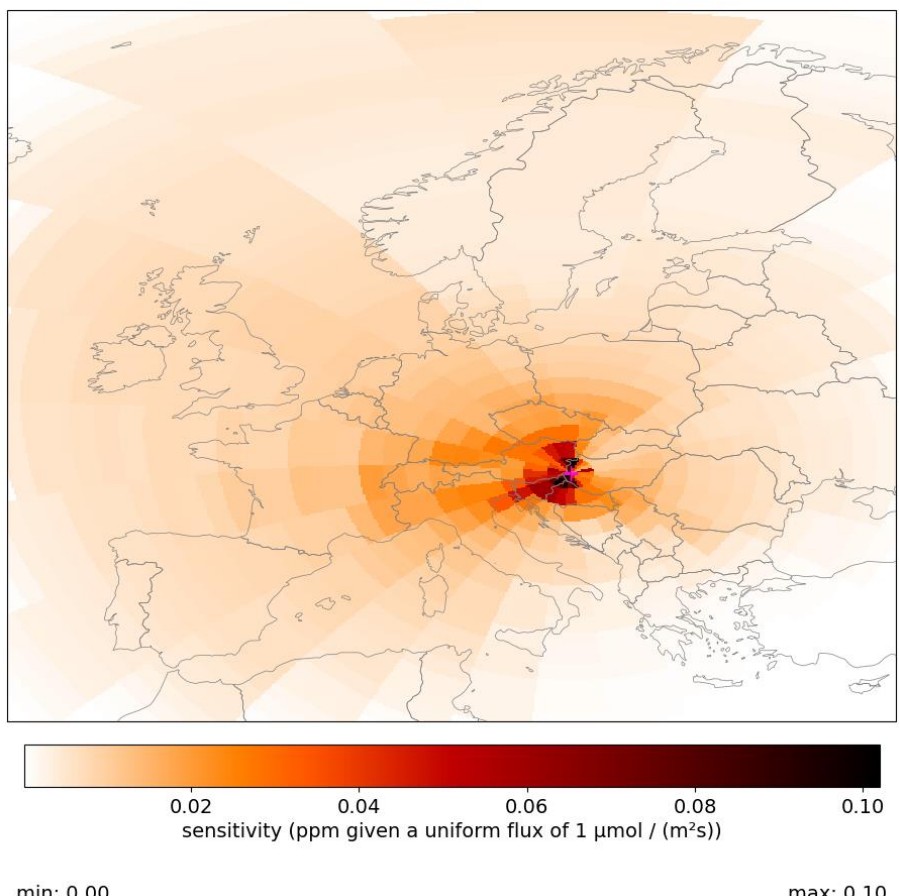

min: 0.00                                                                                      max: 0.10

**Figure 2: The sensitivity area map of the top monitoring level (115 m) at Hegyhátsál tall-tower monitoring site calculated for 2022 (Carbon Portal ICOS RI, 2024).**

The monitoring site has been a member of the WMO Global Atmosphere Watch (GAW) network and of the NOAA Global Cooperative Air Sampling Network since its establishment and joined the pan-European Integrated Carbon Observation System (ICOS) in 2022.

### 2.2 Monitoring system

The monitoring site was established in 1993 as a flask air sampling site for the NOAA Global Cooperative Air Sampling Network (Thoning et al., 1995; NOAA, 2023). In situ measurements of carbon dioxide concentration started in 1994. Since the Hegyhátsál monitoring site was set up within the framework of a US-Hungarian scientific collaboration only one year





after the construction of the first NOAA tall-tower monitoring site in the U.S.A. (North Carolina) (Bakwin et al., 1995), the design of the measuring system at Hegyhátsál (Fig. 3) is similar to that at the US tower (Zhao et al., 1997). The dry air mole fraction of $CO_2$ is measured at 10, 48, 82, and 115 m above the ground. Air is pumped through 3/8 inch diameter tubes (Synflex 1300) to a $CO_2$ analyzer located on the ground floor in an air-conditioned room. Diaphragm pumps are used to continuously draw air through each of the tubes from the four monitoring levels (1994-2005: KNF model UN73MVP, ~2 L

$min^{-1}$; from 2005: KNF model N 811 KN.18, ~6 L $min^{-1}$). After leaving the pump, the air at 40 kPa overpressure enters a glass trap for liquid water that is cooled in a standard household refrigerator, to dry the air to a dew point of 3-4 °C. Liquid water is forced out through an orifice at the bottom of each trap to minimize the loss of $CO_2$ to the liquid phase. The four inlet tubes and the standard gases are connected to a computer-controlled 16-position flow-through rotary selector valve (VICI AG, Valco Europe). The valve selects which intake tube or standard gas is sampled by the analyzer. The valve head is

protected by 7-µm in-line filters. Ambient air continuously flows through the multiport valve so that the system is constantly flushed. The standard gases are shut off when not in use by means of computer-controlled solenoid valves. The air exiting the multiport valve through its common outlet is further dried to a dew point of about -25 °C by passing through a 182-cm-long Nafion drier (Permapure, type MD-110-72P), so that the water vapor interference and dilution effect are <0.1 ppm equivalent $CO_2$ (Zhao et al., 1997). The Nafion drier is purged in a counter flow arrangement using waste sample air that has

been further dried by passage through anhydrous $CaSO_4$ (W.A. Hammond Drierite Co. Ltd.).

Until 2020, $CO_2$ analysis was performed using non-dispersive infrared (NDIR) gas analyzers (1994–2007: Li-Cor Inc. model LI-6251; 2007-2020: Li-Cor Inc. model LI-7000). A constant sample flow rate of 100 $cm^3$ $min^{-1}$ was maintained by a mass flow controller (Tylan, model FC-260). The reference cell of the $CO_2$ analyzer was continuously flushed at a flow rate of 5–10 $cm^3$ $min^{-1}$ with a compressed reference gas of 330–400 µmol $mol^{-1}$ $CO_2$ in synthetic air (Messer Hungarogáz) gradually

following the concentration trend in the atmosphere. Since 2020, a Picarro model G2301 cavity ring-down spectrometer (CRDS) has been used to determine the $CO_2$ content of the air. While the NDIR analyzers required air pressed through the measuring cells, the pressure in the measuring cell (cavity) of the CRDS analyzer is well below the atmospheric one (~185 hPa), which required the partial redesign of the system. CRDS analyzers are absolute instruments, they do not require a continuously flowing reference gas like the NDIR ones. Provision of the low pressure in the analyzer requires a vacuum

pump (Picarro model A2000) after the analyzer in the air stream. The mass flow controller has been set to above the flow requirement of the analyzer, and an overflow bypass has been inserted to prevent the overpressure (generated by the backpressure regulators for the removal of the condensed water) at the inlet of the analyzer. The bypass flow is monitored to prevent any backflow contamination. The modifications to the original system are indicated in Fig. 3 in different colors. This transformation needed the minimum modification of the original design to accommodate the CRDS analyzer until the

monitoring system was rebuilt according to the ICOS recommendations (ICOS RI, 2020) in late 2023.



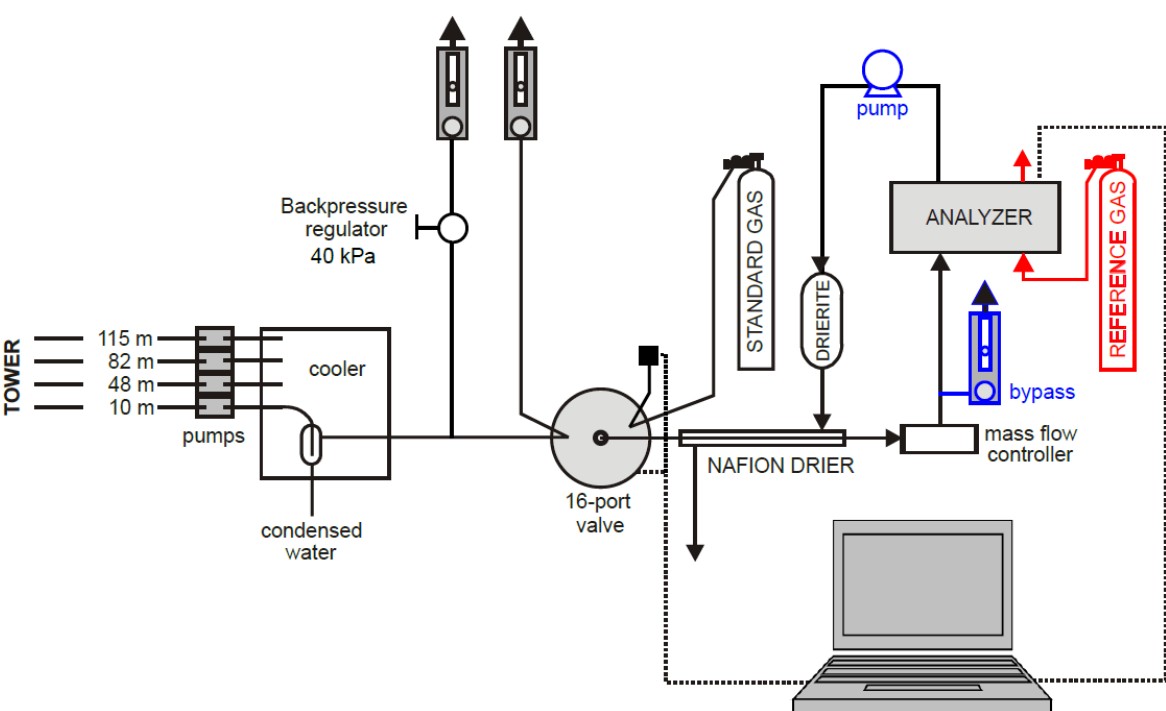

**Figure 3: Schematics of the monitoring system at Hegyhátsál (HUN). Blue parts were added and red parts were removed when the Li-Cor NDIR analyzer was replaced by the Picarro CRDS one in 2020. For clarity, only one sampling line and one standard gas are shown.**

## 2.3 Measurement cycles and calibration

Typically, a single gas analyzer that scans the intakes sequentially is used at a tall tower, multi-elevation monitoring site (see e.g. Bakwin et al., 1998; Thompson et al., 2009; Popa et al., 2010; Sasakawa et al., 2010; Winderlich et al., 2010; Vermeulen et al., 2011; Andrews et al., 2014; Schmidt et al., 2014; Berhanu et al., 2016; Stanley et al., 2018; Conil et al., 2019; ICOS RI, 2020; Lelandais et al., 2022). It involves that the continuous concentration signal is sampled only for discrete short periods at each intake point, which does not allow the perfect reconstruction of the original concentration variation (Andrews et al., 2014; Haszpra and Prácser, 2021). It increases the uncertainty of the calculated hourly averages usually used by the atmospheric inversion and budget models. The uncertainty derived from the non-continuous sampling at the tall-tower sites can be significantly higher than the other terms of the measurement uncertainty (e.g. scale transfer, scale inconsistency, scale drift, instrument noise, etc.) (Haszpra and Prácser, 2021). This type of uncertainty can be reduced by high-frequency sampling at each intake. Obviously, increasing the frequency reduces the time available for flushing the instrument and signal integration at each intake, which also introduces uncertainty into the measured concentrations. Therefore, the decision





on the sampling time/frequency is a compromise. The uncertainty of the hourly average concentration is the highest when the concentration changes rapidly within the hour. Under low-elevation continental conditions, the typical periods are the morning and evening hours when the stable nocturnal boundary layer breaks up/builds up. Today models try to avoid the critical periods and use data only from the afternoon hours when the atmosphere is the best mixed, and the fluctuation of the $CO_2$ concentration is low. This approach loses 75-85 % of the available measurement data. The more precise allocation of

the sources and sinks requires high-resolution models that are able to handle the rapid changes in the planetary boundary layer and that can use not only a small fraction of the available concentration data. Therefore, the measurement protocols should also provide low-uncertainty data for those periods of the day that are not currently used by most of the models. The uncertainty of the measurements cannot be reduced retrospectively when the demand appears.

In the mid-continental planetary boundary layer, the fluctuation of carbon dioxide concentration is high due to the rapidly

changing photosynthesis/respiration of the vegetation and that of the turbulence. The diurnal variation of the height of the planetary boundary layer can also be remarkable. Therefore, we have chosen the shortest realistic sampling time at each air intake to reduce the uncertainty as much as possible. Taking into account the small dead volume of the system to be flushed at each intake change, a two-minute sampling time seemed to be an acceptable compromise (Zhao et al., 1997; Werner et al., 2006). With two minutes of sampling at each of the four sampling elevations, 7-8 subsamples are available for the

calculation of the hourly average concentrations. The 2-minute period consisted of one minute of flushing and one minute of signal integration when the LI-6251 or LI-7000 NDIR analyzer was working in the system. The Picarro model G2301 CRDS analyzer has a larger measuring cell and somewhat lower instrument noise, therefore we changed to 90 s flushing and 30 s integration time when the NDIR analyzers were replaced with the CRDS one. As all the tubes are continuously ventilated, 60-90 s is sufficient for the reasonable flushing of the analyzers. Our experience showed that the difference between the

"true" concentration and the measured concentration at Intake 2 fell below 0.1 µmol mol$^{-1}$ in 35-45 s even if the concentration difference between Intake 1 and Intake 2 was as high as 70 µmol mol$^{-1}$ (Haszpra and Prácser, 2021). During the signal integration, there is no need to wait for any "stabilization" because the atmospheric concentration is never "stable", it varies continuously. When studying a year-long data series with a 5 s temporal resolution it was found that at 82 m above the ground, 38 % of the 40 s periods show statistically significant ($p<0.05$) concentration changes due to the

natural concentration variations (Haszpra and Prácser, 2021).

Four $CO_2$-in-air standards prepared and certified by the WMO Central Calibration Laboratory for $CO_2$ (NOAA, Boulder, Colorado) (Trivett and Kohler, 1999; WMO, 2023) were used to calibrate the measuring system. All data in this paper are reported on the WMO X2019 $CO_2$ scale. The NDIR analyzers are pressure- and temperature-sensitive. The analyzers at Hegyhátsál were not temperature and pressure stabilized, therefore they required frequent calibration to compensate for the

scale drift. Every 32 min, after four 8-minute measuring cycles (4 intakes x 2 minutes), the standard gas with the lowest $CO_2$ mixing ratio was selected and analyzed, and this measurement was termed "zero." After every sixth cycle (every 202 min), a full four-point calibration was carried out, and a new quadratic response function was calculated. The "zero" measurements were used to account for any short-term drift of the analyzer due to changes in ambient pressure or temperature. The "zero"





offsets and response functions were linearly interpolated in time to obtain values suitable for the calculation of $CO_2$ concentrations from the instrument response.

The CRDS analyzers are much less sensitive to the pressure and temperature fluctuations in the environment because they are internally temperature and pressure stabilized. They can operate safely with much less frequent calibration. The Picarro model G2301 CRDS $CO_2$ analyzer, installed in the monitoring system in 2020, was calibrated at monthly intervals against four NOAA certified standards..

## 2.4 Quality assurance and quality control

As it was mentioned in the previous section, for the calibration of the analyzers four $CO_2$-in-air standards, prepared and certified by the WMO Central Calibration Laboratory for $CO_2$ (WMO, 2023) were used. Hegyhátsál is also a site in the NOAA's Global Cooperative Flask Air Sampling Network (NOAA, 2023) (station code: HUN) where every week two glass flasks (2.5 L each) are filled in parallel with ambient air at 96 m elevation on the tower. The samples are analyzed by NOAA

ESRL Carbon Cycle Greenhouse Gases Group (Boulder, Colorado, U.S.A.) for $CO_2$ and several other greenhouse gases. The independent in situ and flask sample measurements provide a continuous quality assurance against any scale drift or other issues, although exact comparability is not possible: i) while the flask air samples are taken at 96 m elevation, the in situ measurements are carried out at 82 m and 115 m; ii) while the flasks contain instantaneous air samples, the in situ measurements are carried out only in every 8th minute at a given sampling elevation, which is further complicated by the

different flow rates and residence times.

The flask air samples are always taken in the early afternoon hours when the atmosphere is the best mixed. The average of the in situ measurements at 82 m and 115 m can fairly well approximate the concentration at the flask sampling elevation (96 m). To reduce the bias due to the not exactly known temporal asynchrony the in situ measurements in the ±20-minute time window around the nominal sampling time are averaged. For the comparison, 1089 flask air sample data were available.

We filtered out those cases (n=22) when the difference between the in situ and the flask measurements was larger than 3σ (presumably caused by extreme concentration variability in the time window, possible sampling or analytical errors, etc.). The mean deviation of the in situ measurements from the flask measurements is -0.08±1.59 µmol mol$^{-1}$, which satisfies the WMO network compatibility goal (0.1 µmol mol$^{-1}$ – WMO, 2020). The relatively large scatter is due to the high temporal variability of the atmospheric concentration at this mid-continental site even at 82-115 m above the ground, and to the

unavoidable spatial and temporal asynchrony of the in situ and flask measurements. Similarly high scatter was also reported by Bakwin et al. (1995) for a tall tower site in North Carolina, U.S.A. Technical issues might also contribute to the scatter and bias. Since the replacement of the LI-6251 analyzer with the more stable LI-7000 in 2007, the bias and the scatter have decreased slightly (-0.04±1.14 µmol mol$^{-1}$).





**2.5 Data selection**

Comparison of the parallel measurements at K-puszta and Hegyhátsál in 1994-1999 showed that the smallest difference between the two sites 220 km away from each other can be observed during the afternoon hours, after 12 h local standard time (LST) (Haszpra, 1999a). This is the natural consequence of the mixing of the atmosphere, which is the most intensive in the early afternoon hours. The study also indicated that, unlike at coastal stations, no clean air sector can be defined at this mid-continental site (Haszpra, 1999b). The air in this geographical basin (Pannonian Basin) is well-mixed, and no directional

difference can be observed far from the significant anthropogenic sources. Therefore, no data selection other than disqualifying the technically false data (instrument malfunction, maintenance, calibration, etc.) is applied. To achieve a higher spatial representativeness, in most of our studies only data from the early afternoon period (12-16 h LST) are used.

**3 Results and discussion**

**3.1 Diurnal cycle of carbon dioxide concentration**

One of the most striking features of the atmospheric carbon dioxide concentration at a mid-continental site surrounded by vegetation is its remarkable diurnal cycle. It is generated by the cyclic photosynthesis and respiration periods of plants, and modulated by the diurnal cycle of the vertical mixing of the atmosphere. Figure 4 shows the monthly average diurnal cycles at the different measurement elevations in different seasons relative to the daily average concentration at the top of the tower. Being the closest to the sources/sinks at the surface (soil, vegetation) the lowest elevation (10 m) shows the highest diurnal

amplitude throughout the year. The maximum diurnal amplitudes can be observed in the middle of the summer at all measurement elevations due to the activity of the vegetation and the high respiration rates during the night. The maximum diurnal amplitudes are 60.5 and 17.2 $\mu$mol mol$^{-1}$ at 10 m and 115 m, respectively, while they are only 6.5 and 1.5 $\mu$mol mol$^{-1}$ in winter (Fig. 5). Atmospheric dynamics also contribute to the high summer diurnal amplitudes. The nighttime boundary layer is the shallowest at this mid-continental site in summer (Fig. S2). The temperature-dependent respiration causes

elevated concentrations in the shallow boundary layer reaching 550-600 $\mu$mol mol$^{-1}$ (hourly average) at 10 m during certain summer nights in the last years.

Although the tower is not very tall, the gradual intrusion of the $CO_2$-enriched nighttime surface layer air into the higher layers is clearly observable (Fig.4), especially in summer. The morning concentration peak at 115 m is delayed by 2-3 hours relative to that at 10 m elevation.











**Figure 4: Monthly mean diurnal variation of $CO_2$ dry air mole fraction at different elevations in different seasons relative to the daily mean at 115 m.**






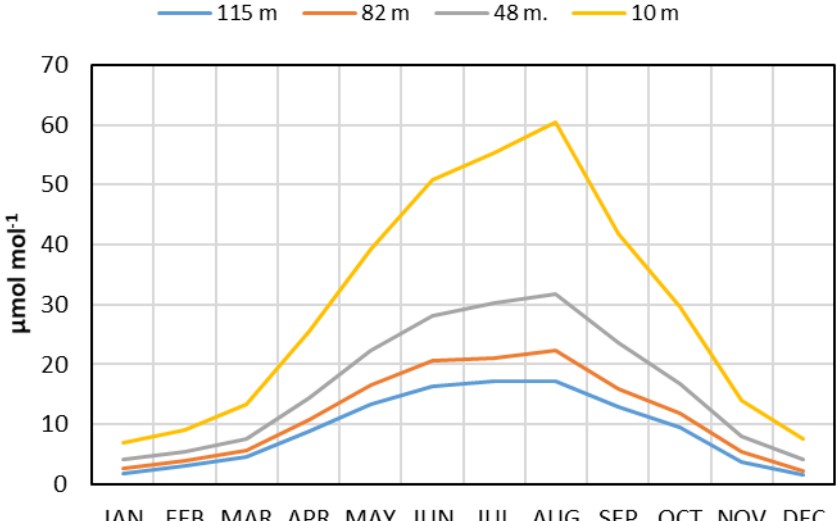

**Figure 5: Seasonal variations of the monthly mean daily amplitudes of $CO_2$ concentration at different elevations.**


The diurnal amplitude at the different measurement elevations has not changed significantly at the usual probability levels (p<0.05) during the past 30 years, however, an upward tendency can be observed throughout the year, especially at the lowest monitoring elevation (10 m). The highest increase ($0.57\pm0.30$ $\mu mol$ $mol^{-1}$ $year^{-1}$, p=0.074) appears in the warmest month of the year, in July (Fig.S3). Comparing the July concentration values from the first and the second halves of the

measurement period (Şen, 2012; Şen, 2017) it seems the high concentrations grow somewhat faster than the lower ones (Fig. 6). According to the ECMWF ERA5 reanalysis, the height of the nighttime boundary layer has not changed during the decades of the measurements around the monitoring site (average of the 4 grid points surrounding the site). Based on a shorter data series Perez et al. (2020) also noticed a higher-than-average trend for the 90-percentile concentration values. The high concentrations are typical in the nighttime surface layer due to the respiration of the ecological systems. The intensity of

respiration is typically an exponential-like function of the temperature (see e.g. Meyer et al., 2018). As the nighttime minimum temperature at Hegyhátsál is statistically significantly increasing (Fig. S4) the likely reason for the upward tendency in the diurnal amplitude is the increasing respiration due to the significant increase in temperature.

Sources and sinks at the surface cause a vertical concentration gradient even in the early afternoon hours when the atmosphere is most mixed. Due to the seasonally varying source/sink yield, the early afternoon concentration gradient is

positive (concentration increases with elevation) from late March to October, and negative during the winter half-year (Fig. 7). According to the airborne measurement campaigns in 2006-2008 (Haszpra et al., 2015), the top of the tower underestimates the mean planetary boundary layer concentration by 0.9 $\mu mol$ $mol^{-1}$ in summer afternoons, and overestimates





it by 1.2 µmol mol$^{-1}$ in winter (median values). The interquartile ranges are [0.16 – 1.54] and [-0.,2 – 3.04] µmol mol$^{-1}$, respectively. These results may be informative for those using models with coarse vertical resolution.


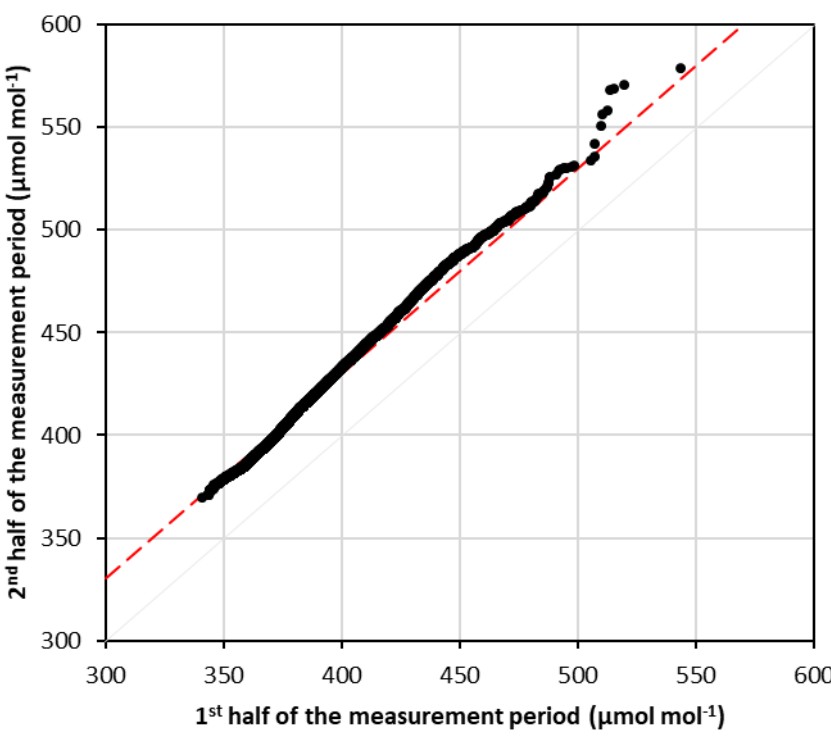

**Figure 6: Şen-diagram showing the ordered pairs of the hourly average CO$_2$ concentrations measured at 10 m in July in the first half (x) and the second half (y) of the measurement period. The shift of the red dashed line from the 1:1 line gives half of the concentration increase during the measurement period. The deviation of the data points from the red dashed line shows how the trend of the given concentration range deviates from the overall trend. The upward deviations of the high-concentration data points from the red dashed line mean that the higher concentrations increased more than the lower concentrations. The overall trend is dominated by the x<400 µmol mol$^{-1}$ data points representing 78.5 % of the total data series (n=8344).**






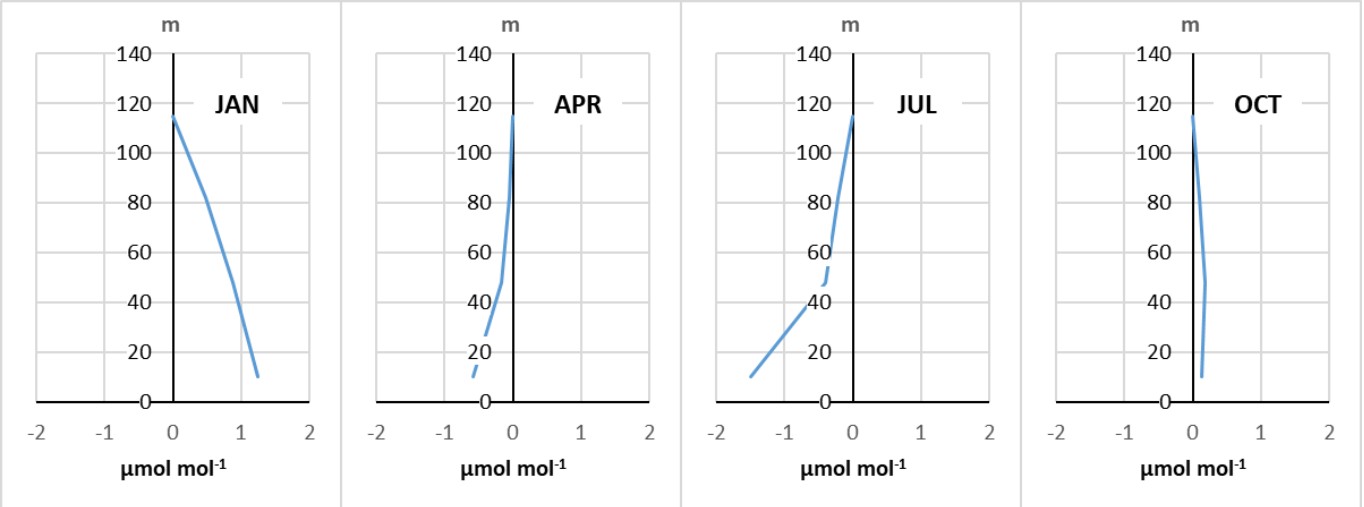

**Figure 7: Monthly mean early afternoon (12-16 h LST) vertical concentration gradients in different seasons relative to the concentration at 115 m.**


## 3.2 Seasonal variations of carbon dioxide concentration

In addition to the diurnal variation, the cyclic activity of the temperate zone vegetation also generates a remarkable seasonal variation in the atmospheric carbon dioxide concentration. For its evaluation, the detrended concentration time series was used. The data series was detrended by applying the widely used CCGCRV software (Schmidt et al., 2003; Zhao and Zeng,

2014; Fang et al., 2015; Zhu and Yoshikawa-Inoue, 2015; Piao et al., 2018; Curcoll et al., 2019; Reum et al., 2019; Lin et al., 2020; Wang et al., 2020; Xia et al., 2020; Resovsky et al., 2021; Tiemoko et al., 2021 – and others) developed by Thoning et al. (1989), and publicly available at https://gml.noaa.gov/ccgg/mbl/crvfit/crvfit.html (last accessed 13 October 2023). The mean seasonal cycles for 10 m and 115 m can be seen in Fig. 8. The figure shows the seasonal cycles calculated from both the daily average concentrations and the early afternoon (12-16 h LST) measurements, which are more spatially

representative. There is a significant difference between the amplitudes of the seasonal cycles of the whole-day data and the early afternoon data. The reason for this is that the summer daily means are much higher than the early afternoon values due to the intensive respiration and the accumulation of carbon dioxide respired to the shallow nighttime boundary layer, and it compresses the range of the seasonal variation. The mean seasonal amplitude based on the early afternoon (12-16 h LST) measurements is 25.8 µmol mol$^{-1}$ at 115 m, and it is 29.3 µmol mol$^{-1}$ at 10 m where the influence of the overlying vegetation

is more pronounced. The high seasonal amplitude is the consequence of the cyclic behavior of the ecological systems surrounding the monitoring station. Comparable high seasonal amplitudes are reported only from a few sites (Popa et al.,



2010; Vermeulen et al., 2011; Liu et al., 2014; Conil et al., 2019; Curcoll et al., 2019) the majority of which are also located in a mid-continental environment.

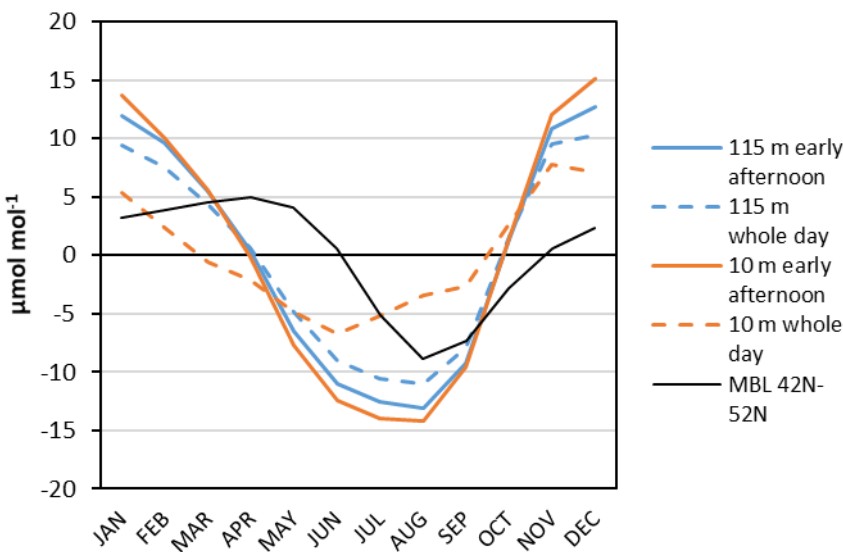


**Figure 8: The mean seasonal variations at Hegyhátsál at 10 m and 115 m elevations based on the early afternoon (12-16 h LST) measurements and the whole day measurements. The figure also shows the mean seasonal variation of the marine boundary layer reference concentrations characteristic for the 42°N-52°N latitudinal band (MBL 42N-52N).**


Figure 8 also presents the seasonal variation of the marine boundary layer reference concentration (Lan et al., 2023b) for the 42°N-52°N latitudinal belt where Hegyhátsál station is located. The shape of its annual cycle significantly differs from those measured at our mid-continental site. In the marine boundary layer, the maximum of the annual cycle forms in April when the ecological systems in the northern temperate zone turn from a net $CO_2$ source to a net $CO_2$ sink. At Hegyhátsál, unlike in

the marine boundary layer, the $CO_2$ concentration reaches its annual maximum at the end of the year, long before the vegetation becomes a net $CO_2$ sink. Such a big difference cannot be explained by horizontal transport from regions with depleted $CO_2$ concentrations. The main reason is the changing dynamics of the regional atmosphere. From January, following the increasing insolation after the winter solstice, the vertical mixing is rapidly getting more vigorous (see Fig. S2) mixing more and more relatively clean free tropospheric air into the $CO_2$-enriched boundary layer. This process

overcompensates the contribution of the surface that is still a net $CO_2$ source at that time of the year. A simple box model used in our earlier work (Haszpra and Barcza, 2010) proved that the changing vertical mixing can result in the observed reduction in the late winter/early spring surface layer concentrations. Concentration maximums preceding the net uptake period of the regional vegetation are also be observed at other monitoring sites (Levin et al., 1995; Davis et al., 2003; Kozlova et al., 2008; Popa et al., 2010; Schmidt et al., 2014;-Belikov et al., 2019; Conil et al., 2019; Curcoll et al., 2019;





Lelandais et al., 2022; Jiang et al., 2023) but to the author's knowledge, the role of the changes in the dynamics of the atmosphere has not yet been studied in details.

The length of the summer season, when the $CO_2$ concentration is negative relative to the annual average ($CO_2$ deficit season) has increased. Defining its beginning by the day of the year when the $CO_2$ concentration becomes negative relative to the annual average (spring zero-crossing), the beginning of the summer season is getting earlier by $-0.38\pm0.16$ day year$^{-1}$

(p=0.024) (Fig. 9). Although this linear trend over the entire measurement period is statistically significant at an acceptable level of confidence, the data suggest faster progress at the beginning of the period, until about the end of the 2000s, and a stagnation thereafter. The linear trend of the spring zero-crossings between 1995 and 2011 is $-0.96\pm0.41$ day year$^{-1}$ (p=0.033), with an average date of the zero-crossing at the 110.7th day of the year (DOY), while from 2011 through 2023 it is $0.08\pm0.32$ day year$^{-1}$ (p=0.81), with an average of 106.7 DOY.


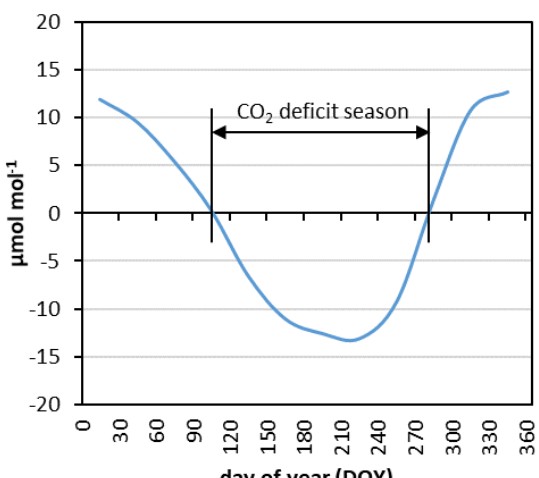

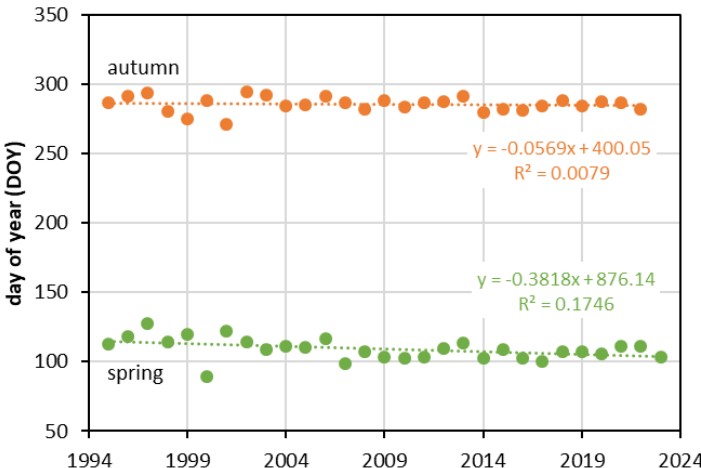

**Figure 9: Temporal variations of the beginning and end of the $CO_2$ deficit season, the spring and autumn zero-crossings.**

Not forgetting about the contribution of the dynamics of the atmosphere mentioned above, the spring drawdown of the atmospheric carbon dioxide concentration is essentially caused by the increasing activity of the biosphere, and the date of the spring zero-crossing is closely related to the start of the growing season of the vegetation in the temperate and arctic regions. Numerous phenological studies show the shift in the timing of the spring leaf unfolding, and the increasing greening of the terrestrial vegetation in these geographical zones (Schwartz et al., 2006; Peñuelas et al., 2009; Piao et al., 2019; Wang et al.,

2019; Chen and Yang, 2020; Piao et al., 2020; Vitasse et al., 2022; Rahmati et al., 2023). The phenomenon has multiple drivers, but climate warming is the main contributor. Despite the steady rise of the temperature, the phenological and NDVI observations indicate that the advancement of the start of the growing season slowed down in the last decade (Piao et al.,





2017; Rahmati et al., 2023). Our atmospheric carbon dioxide measurements, the changing trend of the spring zero-crossing also support the decreasing temperature control over the start of the growing season. The possible explanations for this phenomenon are the reduced chilling during the dormant period and the emerging light limitation (Piao et al., 2017) that prevents the even earlier start of the growing season despite the otherwise increasingly favorable conditions.

In a large part of Europe, the timing of the end of the growing season does not show a significant trend (Rahmati et al., 2023). The autumn zero-crossing does not have any significant trend at Hegyhátsál either (Fig. 9). The end of the growing season is determined by multiple factors, and it only weakly correlates with temperature (Wang et al., 2019). The warming summer may lead to increased atmospheric water demand, reduced soil moisture, and the earlier start of the senescence (Rahmati et al., 2023; Zhang et al., 2023). In Central Europe, where large regions are covered by agricultural fields, the summer/early autumn harvest of crops may also contribute to the fact that the net carbon uptake period is not longer despite the prolonged warming, which is in principle favorable for vegetation.

The earlier start of the growing season, the general warming, and the increasing $CO_2$ fertilization make the $CO_2$ uptake of the vegetation more intensive, which is reflected in the increasing NDVI values. The warming also increases respiration, and in principle, the more intensive $CO_2$ uptake and the more intensive respiration result in increasing seasonal amplitude in the atmospheric $CO_2$ concentration. The increasing amplitude was observed at several monitoring sites (Keeling et al., 1996; Graven et al., 2013; Forkel et al., 2016; Liptak et al., 2017; Piao et al., 2018; Jin et al., 2022). However, the increasing trend slowed down significantly around the turn of the millennium likely due to the increasing temperature-increased summer respiration, evaporative demand, and water stress (Wang et al., 2018; Yin et al., 2018; Wang et al., 2020; Yu et al., 2021; Rahmati et al., 2023). The measurements started at Hegyhátsál in 1994 do not show any statistically significant trend in the seasonal amplitude at the commonly applied probability levels (Fig. 10) but it seems as if they indicated a minor negative tendency. Mathematically, this is the result of the increased summer minima with a minor contribution of the reduced winter peaks (Fig. 10). Although it might be the first sign of the consequences of the European emission control measures (Mcgrath et al., 2023) decreasing the winter peaks, and the decreasing summer biospheric uptake due to the declining environmental conditions including more frequent heatwaves and prolonged droughts (Bastos et al., 2020; Ramonet et al., 2020; van der Woude et al., 2023), it cannot be stated with certainty at this stage, however, it is worth keeping an eye on the progress.



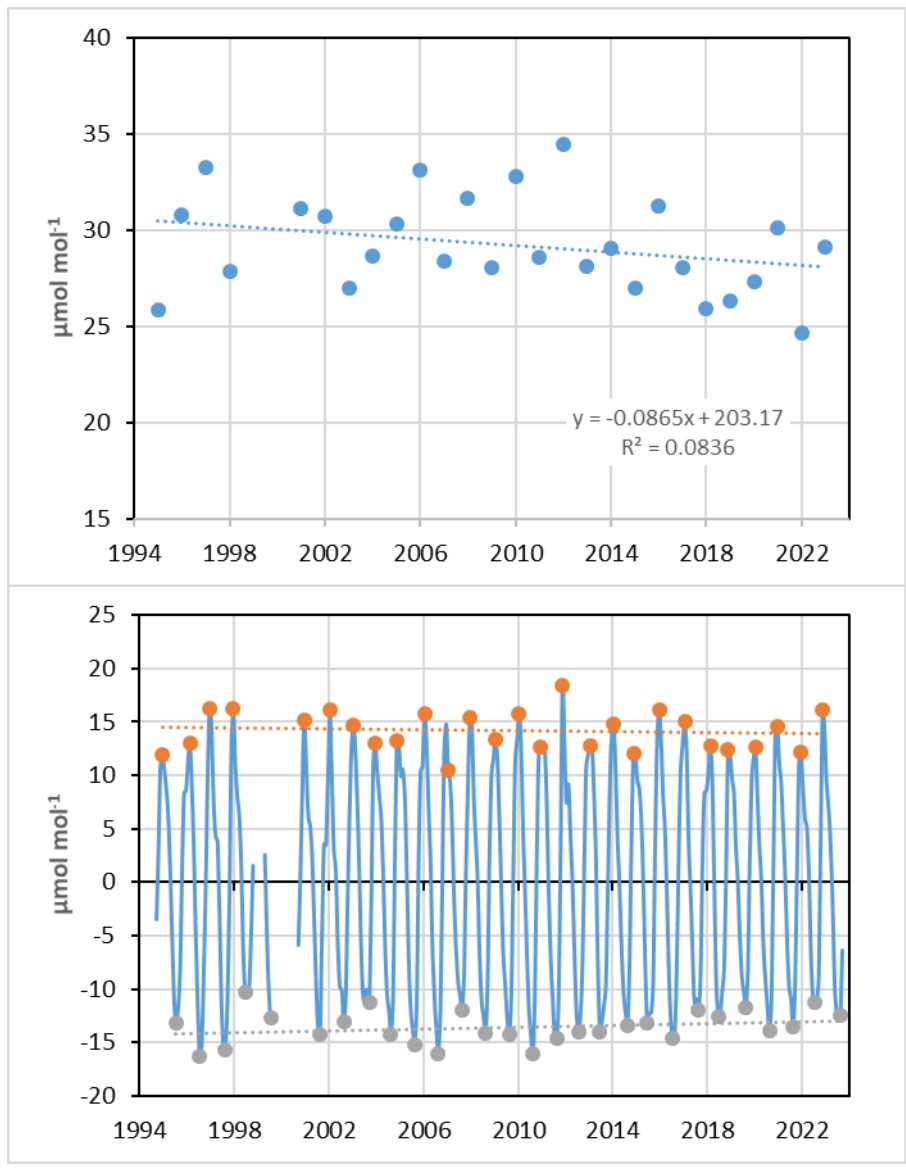

**Figure 10: Temporal variation of the seasonal amplitude at Hegyhátsál at 115 m elevation (upper panel), and the tendencies in summer minimums and winter maximums. 1999-2000 was not taken into account in the trend estimations due to the repeated and prolonged data gaps caused by technical problems.**





### 3.3 Long-term changes in the atmospheric CO₂ concentration

Figure 11 shows the smoothed temporal variation and long-term trend calculated by the CCGCRV software (Thoning et al., 1989). The data indicate a permanent increase in the concentration with an overall growth rate of 2.20 µmol mol$^{-1}$ year$^{-1}$. This is close to what the marine boundary layer reference concentration (Lan et al., 2023b) gives for the 42°N-52°N latitude band (2.16 µmol mol$^{-1}$ year$^{-1}$) and to the overall global growth rate of 2.09 µmol mol$^{-1}$ year$^{-1}$ (Lan et al., 2023a). Not considering the wide seasonal variation, the data from Hegyhátsál show a 3.90±0.83 µmol mol$^{-1}$ offset relative to the marine

boundary layer reference concentration (Fig. 11). The offset is presumably caused by the anthropogenic emissions in Europe.

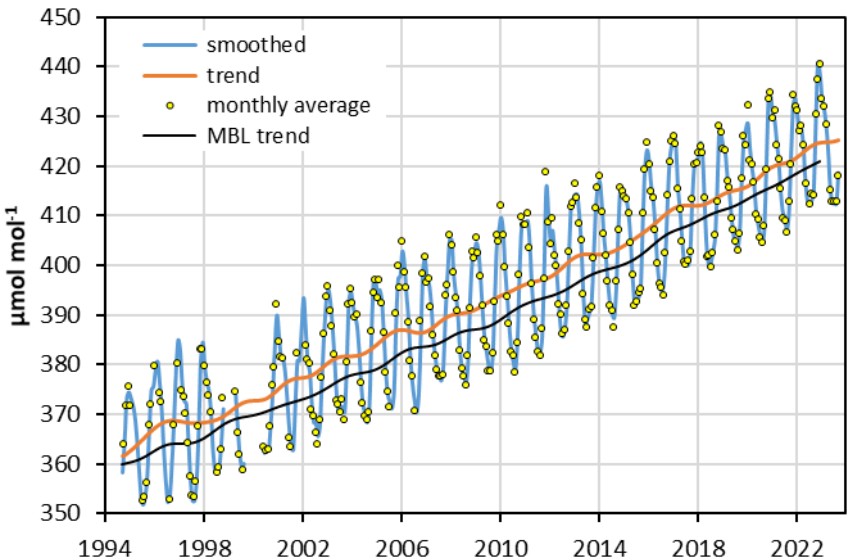

**Figure 11:** **The temporal variation of CO₂ concentration at 115 m elevation at Hegyhátsál showing the monthly averages, the fitted**
**smoothed curve, and the trend, as well as the trend of the marine boundary layer reference concentration in the 42°N-52°N**
**latitudinal bend.**

At Hegyhátsál, the growth rate, the derivative of the time-dependent trend varies between -1.35 and +4.94 µmol mol$^{-1}$ year$^{-1}$ (Fig. 12). Both global and regional anthropogenic emissions vary in much narrower ranges. As it was first noticed by
Bacastow studying the measurements from Mauna Loa Observatory (Hawaii) and the South Pole, the growth rate of atmospheric carbon dioxide concentration is strongly modulated by the El Niño/Southern Oscillation (ENSO) phenomenon (Bacastow, 1976). ENSO is an irregularly periodic variation in winds, pressure, and sea surface temperatures over the tropical Pacific Ocean. In general, its warm phase (also called El Niño) is associated with positive temperature and negative precipitation anomalies, especially in the tropics. The high temperature and prolonged drought conditions reduce the gross
primary production of the vegetation and increase fire activity, which reduces the net carbon uptake (Patra et al., 2005; Zeng





et al., 2005; Kim et al., 2016; Bastos et al., 2018; Rödenbeck et al., 2018). The negative phase of ENSO causes the opposite effects. The ENSO-associated land-atmosphere $CO_2$ exchange variability can be as high as 5 PgC year$^{-1}$ (Zeng et al., 2005), comparable to the atmospheric fraction of current anthropogenic $CO_2$ emission. The ENSO-associated (dominantly) tropical land uptake fluctuations lead to large variations in atmospheric $CO_2$ growth rates that lag ENSO by some time.


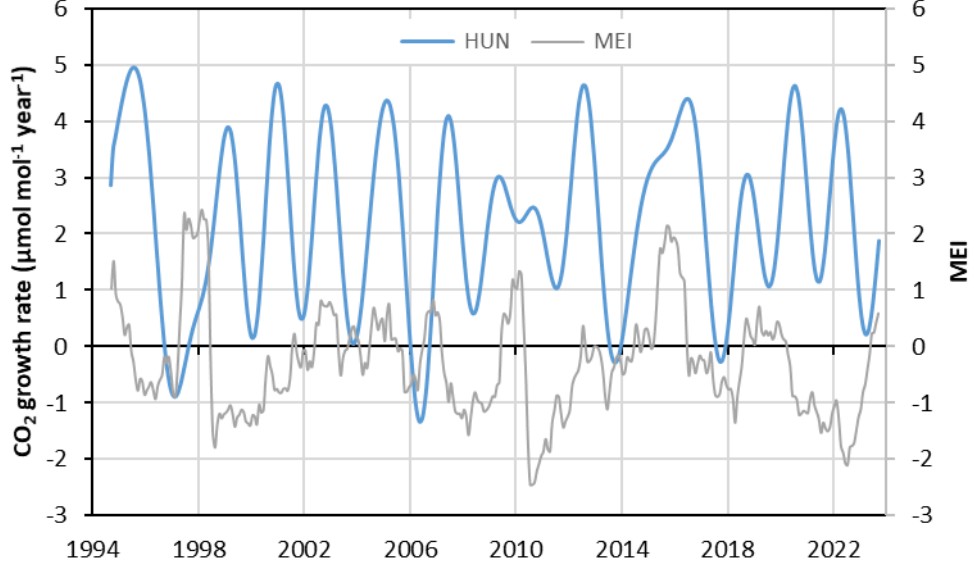

**Figure 12: The temporal variation of the growth rate of $CO_2$ concentration at 115 m elevation at Hegyhátsál and that of the Multivariate ENSO index (MEI).**


Mauna Loa Observatory is located almost in the middle of the Pacific Basin, in Hawaii (20° N), the closest to the ENSO phenomenon. Chylek et al. (2018) reported a lag time of 5.2±2.7 months in the correlation between ENSO and $CO_2$ growth rate. Barring Head, New Zealand, also in the Pacific Basin but at 41° S, also reported a 5-month lag (Stephens et al., 2013). The ENSO relationship is less robust in Europe. Cundari et al. (1995) calculated the correlation between ENSO and the $CO_2$

growth rate measured at Mt. Cimone, Italy, and obtained the maximum correlation coefficient (0.6) with a 7-8-month lag-time. Apadula et al. (2019) reported a 4-month lag at Plateau Rose, Italy, while Artuso et al. (2009) calculated 9 months for Lampedusa, the Italian island in the Mediterranean Sea. Using measurement data from 43 monitoring stations Das et al. (2022) investigated how the ENSO signal in the atmospheric $CO_2$ growth rate propagates meridionally, and how the lag time changes with latitude. They also found significant scatter in the northern extratropics ranging from 0 to 12 months.

ENSO is characterized by different indices. The most commonly used are the Southern Oscillation Index (SOI) and the Multivariate ENSO Index (MEI) (https://www.weather.gov/fwd/indices). For the analysis of the Hungarian measurements, the values of the Multivariate ENSO Index downloaded from the NOAA Physical Sciences Laboratory





(https://psl.noaa.gov/enso/mei/ - accessed on 6 December 2023) were used (Fig. 13). The maximum correlation (r=0.45; p<0.01) was calculated when the growth rate values were delayed by 6 months relative to the MEI ones. The 7-month delay

gave practically the same value (r=0.44). The latitudinally representative Marine Boundary Layer reference data series showed the highest correlation (0.53) for the same period (1994-2023) with a 7-month delay (Fig. 13).

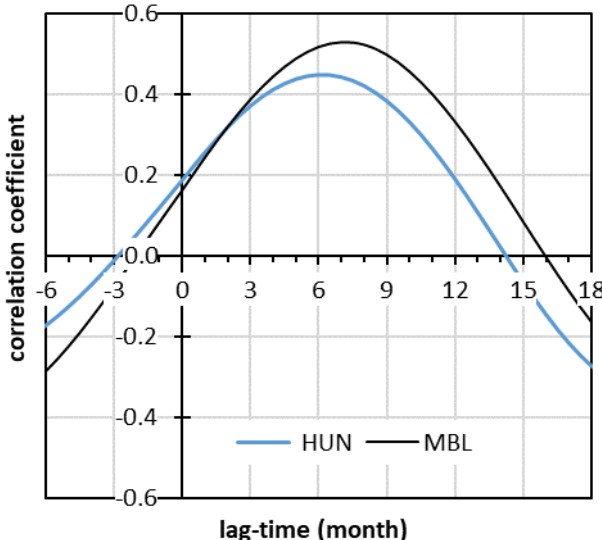

**Figure 13: Correlation coefficient between the CO₂ growth rate and the Multivariate ENSO Index at Hegyhátsál and in the marine boundary layer as the functions of the lag-time.**

## 4 Summary

In this study, we analyzed the 3-decade-long data series from Hegyhátsál, a Central European monitoring site belonging to

the WMO GAW, NOAA, and ICOS networks. The data series is unique in the sense that most of the monitoring sites with similar characteristics (low-elevation mid-continental sites surrounded by active vegetation) have much shorter data series yet. The measurements are widely used in atmospheric inversion and budget models as a part of global datasets. In their use, the remarkable, seasonally varying diurnal variation of the concentration should be taken into account. The long-term trend follows that in the global background atmosphere with a few $\mu mol\ mol^{-1}$ positive offset, which is the contribution of the

European net emissions. Although the monitoring site is far from the Pacific Basin and several factors are making the data series noisier than at the global background stations, the effect of the Southern Oscillation is clearly detectable in the data series.

**Acknowledgments**

The monitoring program at Hegyhátsál was initiated by Ernő Mészáros (Hungarian Meteorological Service) and Pieter Tans
(NOAA), with contribution by Peter Bakwin (NOAA), and supported by Antenna Hungaria Corp., the owner of the
telecommunication tower used. During the initial years, the monitoring program was funded by the U.S-Hungarian Scientific
and Technological Joint Fund, later by several European Union (AEROCARB, CHIOTTO, CarboEurope, CarboEurope-IP,
Carbon-Pro, IMECC, InGOS, RINGO) and Hungarian research projects (OTKA). The monitoring site was operated by the
Hungarian Meteorological Service until 2020 when it was taken over by the Institute for Nuclear Research (Hungary). The
compilation of the present review study was supported by the Hungarian National Research, Development and Innovation
Office (grant OTKA K141839). The author thanks Kirk Thoning (NOAA) for the CCGCRV software used in the data
evaluation and Ida Storm (ICOS Carbon Portal) for the footprint calculations. The author also thanks the staff of the NOAA
ESRL Global Monitoring Laboratory for the long-term cooperation and analyses of the air samples taken at Hegyhátsál.

**Data availability**

The carbon dioxide concentration data measured at Hegyhátsál (HUN) are publicly available at the WMO World Data
Centre for Greenhouse Gases (https://gaw.kishou.go.jp/), and as part of NOAA's ObsPack data product
(https://gml.noaa.gov/ccgg/obspack/).

**Author contributions**

LH has been the principal investigator of the $CO_2$ measurements at Hegyhátsál since the construction of the monitoring
station.

**Competing interests**

The author has declared no competing interests.

**Financial support**

The compilation of the present review study was supported by the Hungarian National Research, Development and
Innovation Office (grant OTKA K141839).





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
