# Peer review of "Multi-decadal atmospheric carbon dioxide measurements in Central Europe, Hungary"

_Atmospheric Measurement Techniques, 2024_

## Author Comment (AC1)

**Authors' response to Referee#1**

First of all, we would like to thank the Referee for his/her positive evaluation of our manuscript and would like to thank him/her for the comments and suggestions, which have helped us to improve the manuscript. Below you will find our detailed, point-by-point responses to the comments and suggestions. Responses are given in blue.

General:

The author of the manuscript presents and discusses long-term CO2 measurements at a Central European station in Hungary. I enjoyed reading it. The manuscript is nicely written and organized and can be considered for publication with only minor changes outlined below.

Thank you for the positive evaluation!

Minor points:

L 10 …gives the technical description and its changes over time…

The addition was accepted and inserted in the sentence.

L 10 …physical environment
What does this mean? what about other conditions like to agricultural, biological situations?

„Physical" was used as an „umbrella" term for all environmental factors potentially affecting the measurements (climate, human activities, vegetation, soil characteristics, relief, etc.). To avoid the misinterpretation the attribute „physical" has been omitted.

L 16-17 You may state by how many days the growing period has lengthened over the entire measurement period.

A sentence has been added that gives the increase in the length of the $CO_2$-deficit season during the measurement period (1995-2022: 9.0±6.1 days).

L 63 yes, but what do you tell us about this? In the western European part, there are a couple of more southern stations!

The prevailing wind in the non-Arctic part of Europe blows from the west and carries the emitted $CO_2$ to the east. The southwestern monitoring stations are less helpful than the eastern ones in assessing the emissions in Western Europe, which gives their importance. In Finland, there are monitoring sites geographically east of HUN but they are characterized by the Arctic circulation pattern.

L 68-69 I guess it is enough if you state this in section 2.2.

The sentences have been moved to the end of Section 2.2.

L 74 air intakes...

How many and at which levels? This information can be given already here. At least give the number of air intakes or move the sentence of line 131-132 to here.

This section describes the monitoring site and its characteristics. The characteristics of the monitoring system, including the number and heights of the intakes are given in the next section (2.2 Monitoring system). Although we could also mention the number of intakes and their heights here, this would break the structure of the section and cause the duplication of the information. In the description of the monitoring system, the number and locations of the intakes have to be given for completeness anyway.

L 94-96 These numbers are very low, what about the uncertainty of these values? The values tell us that there is no influence during winter from local emissions.

Thank you very much for the question! Reading the question it was realized that the average excess is not the right term to characterize the local influence on the measurements. It makes little sense because the frequency distribution of the excess $CO_2$ is very far from normal. Instead, it is said that during 83.2 % of the studied period, the local emissions from the nearby village did not reach the sampling intake at all because either the emission sources were leeward to the monitoring site or an inversion layer prevented the intake height from contamination. The excess $CO_2$ derived from the local anthropogenic emissions exceeded 0.04 µmol mol$^{-1}$ only during 0.09 % of the studied time. The sentence was rewritten accordingly.

L 108-109 Reference: I could not find this information. You may consider a reference directly to the original publication where these classification are defined.

The referenced WDCGG web page lists the metadata of all monitoring stations, including their climate classification. You should scroll down to HUN (Hegyhátsál) and you will find the climate classification of the station. Unfortunately, the structure of the WDCGG website does not allow a more direct link to the data of the monitoring site. For easier access to the information the revised manuscript refers to the GAWSIS database.

L 153-154 Also the NDIR systems can be run as absolute measurement devices. It is a question of calibration.

Yes, they can. We used the NDIR analyzers in relative mode to achieve higher sensitivity.

Fig. 3 Resolution of the graph should be increased!

We will contact AMT technical staff about this issue. The figure in the copy of the manuscript downloaded from AMT is sharp and clearly legible. However, if there is any problem with the resolution of the figure we will do our best to provide a higher quality.

L 201 What are the intake 1 and intake 2?

We wanted to show that even if the concentration difference between two intakes (hypothetical Intake 1 and Intake 2) is as high as 70 µmol mol$^{-1}$, the deviation from the true concentration falls below 0.1 µmol mol$^{-1}$ within 35-45 s after switching from the one intake to the other. The sentence has been reworded for clarity.

L 201-203 This is somewhat misleading as in this case, you could use all 2-minute values? Of course, this does not make sense as it is required to flush the Picarro measurement cell.

The sentence has been rephrased for easier understanding

L 218 Calibration: this is rather infrequent with monthly calibration

Since the compilation of the manuscript, we performed a 6-month test period in cooperation with ICOS Atmospheric Thematic Centre. As a result of the rigorous tests, it was agreed that monthly calibration is enough to provide high-quality data for the ICOS monitoring network.

L 220 What about the target tank concentration measurements? Or stability of standard gas raw measurements

No target gas measurement was included in the measurement protocol. In the case of the NDIR analyzers, the frequent "zeroing" could be considered as some kind of target measurement but in fact, those measurements were used for drift compensation. The continuous drift compensation makes it difficult to evaluate the stability of the raw standard gas measurements. Taking into account the very low drift of the Picarro analyzer, originally no target measurements were planned. It was introduced only in late 2023 to comply with the ICOS requirements.

Fig. 4 I guess you have averaged all days of the corresponding month of the complete data series. It would be good to add for each height uncertainty range by light shadowing using the same color.

Unfortunately, this is technically not feasible. Hegyhátsál is a mid-continental monitoring site where the concentration varies considerably. The scatter of the data is high. Even the 1-sigma uncertainty ranges would completely overlap each other during the daytime and partly during the other times of the day.

Fig. 5 The same here, add the range of variation by color shadowing

See the response to your comment regarding Fig. 4.

L 311-312 You might move this sentence after the next sentence and start with: Furthermore, according to...

The two sentences have been swapped as suggested.

L 321-323 How was this calculated?

The mean planetary boundary layer (PBL) concentrations were estimated from the vertical concentration profile along the tower (10-115 m) and what was measured from the aircraft (from ~200 m to the top of PBL). The estimated mean concentrations were compared with those measured at the top of the tower (115 m above ground). Details are given in the referenced paper.

Fig. 6 The shift of the red dashed line….
if the increase is uniform

The sentence has been completed.

Fig. 8 Add see text for reference

The reference has been added.

Fig. 9 Which level have you used here? The lowest I guess. Anyhow write it.

For consistency with the trend calculations, the measurements from the top of the tower (115 m) were used for the evaluation. Now, this is explicitly stated in the caption of Fig. 9.

L 415 NDVI values …add reference

The reference has been added.

Fig. 10 Define summer and winter by specifying the summer (months of...) as well as winter (months of ...)

The caption of Fig. 10 has been completed by defining the summer and winter periods.

L 439 This number could be checked by radiocarbon measurements in comparison with those of a marine boundary layer site.

In the revised manuscript we refer to the $^{14}CO_2$ measurements performed at Hegyhátsál in 2008-2014, which showed a 1-6 ppm seasonally varying fossil fuel excess $CO_2$ relative to the Jungfraujoch European high-mountain baseline station.

L 451 Ocean phenomenon, how does it influence the terrestrial site as strongly as the marine boundary layer? (Fig. 13)

Although El Niño is an oceanic phenomenon, it dominantly influences the atmospheric $CO_2$ concentration through the terrestrial biosphere. The droughts and wildfires it causes add more $CO_2$ to the atmosphere than the extra release from the warming ocean.

Fig. 13 Move the y-axis to the left

y-axis labels have been moved to the left side of the figure.

---

## Author Comment (AC2)

**Authors' response to Referee#2**

First of all, we would like to thank the Referee for his/her positive evaluation of our manuscript and would like to thank him/her for the comments and suggestions, which have helped us to improve the manuscript. Below you will find our detailed, point-by-point responses to the comments and suggestions. Responses are given in blue.

General comment

This paper presents 30 years of measurements of atmospheric carbon dioxide from a tall tower in Hungary. The Hegyhátsál station and its $CO_2$ measurements haveal ready been well characterised in numerous previous publications and this paper adds an overarching view for the continuous long time series from 1994 to 2023 before joining the Integrated Carbon Observation System (ICOS) network. The changes in instrumentation and measurement setups during that period are presented and sampling uncertainties evaluated. Using the 30 years' worth of data, seasonal trends and features of the vertical gradient of $CO_2$ (10-115 m above ground) are investigated and put into a wider context. Changes such as the $CO_2$ seasonal trends and anomalies are tested for their statistical significance and explanations considered. Long-term trends and changes in growth rates are shown and connections are made to the El Niño-Southern Oscillation (ENSO).

The paper makes a valuable contribution to the body of work investigating and interpreting long time series of in situ carbon dioxide measurements from ground-based monitoring stations. Publication is recommended once the minor issues listed below have been addressed.

Specific questions/issues

1 Introduction

Line 23: "measurements were not convincing" – it should be clarified to whom they were not convincing, e.g. the scientific community at the time, or rather from current view point or else. Also, the mentioned technical and representativity problems should be briefly explained, named or outlined.

The trend shown by Callender was not convincing because of the uncertainties caused by the large scatter of the data, the different measurement methods used, the different sampling protocols, and the different characteristics of the sampling locations. The sentence in line 23 of the original manuscript has been completed by the above explanation.

Line 27: for context, consider naming the major measurement networks in which the monitoring stations were established, e.g. NOAA, GAW, etc. and/or add reference to World Data Centre for Greenhouse Gases.

In the revised manuscript, we mention the WMO BAPMoN/GAW network and the NOAA Global Cooperative Air Sampling Network. WDCGG is mentioned as the data source.

Line 28/29: consider adding the first or longest running station as an example each for the list of "arctic regions, high mountain peaks, mid-oceanic islands".

It is a general sentence saying that the first monitoring sites were set up in very isolated remote places. In the parentheses, which can be omitted without changing the meaning of the sentence, give some examples of typical "very isolated remote places". Giving examples of the examples in further embedded parentheses would make the sentence too complex and difficult to follow. Nevertheless, if the Referee insists on seeing example stations we can add them in a second revision.

Line 31: "they could not provide detailed information", give an example of what is meant here with "detailed information"

The sentence has been completed by the biosphere-climate interaction as an example for which the remote sites could not give detailed information.

Line 38: give a brief explanation what is meant by tall tower, e.g. "towers of up to 100s of metres high"

The definition of tall towers as 100+ meter tall structures is given in the revised sentence.

Line 39: influence by local vegetation: any type of localised influence should be relevant here, including local anthropogenic sources such as e.g. from industrial activity. Consider rephrasing.

The sentence has been rephrased to cover all local natural or anthropogenic sources/sinks.

Line 43-44: Five years of parallel measurements of K-puszta and Hegyhátsál: It would have been interesting to include the period of parallel measurements in this manuscript to extend the time series from the region, and add information to the spatial representativity of the measurements. If such an analysis already exists, please provide a citation or consider adding a short section that includes the measurements at K-puszta.

A paper on this topic was published at the end of the parallel measurements (Haszpra, 1999a), which is referred to in Section 2.5 of the original manuscript. A comparison and possible combination of the two data series would go far beyond the scope of the present manuscript. The discussion of the effects of the different instrumentation, sampling protocol, and environment (different soil types, land cover, etc.) would blow up the paper, therefore, we did not want to go in that direction.

Line 50: determination of the carbon budget of the atmosphere: consider adding a reference, such as e.g. the Global Carbon project https://www.globalcarbonproject.org/index.htm

In this general introductory section, we did not want to give a long list of $CO_2$ inversion models. However, we have included the Global Carbon Project in the revised manuscript as a kind of umbrella project in this field.

Line 58: model results depends on coverage: is temporal or spatial coverage meant here? What role does the quality of measurements play?

The emphasis is on spatial coverage because a monitoring site can only be considered to exist if it produces data, i.e. the temporal coverage is given. The sentence referred to has been rephrased. The higher the uncertainty of the measurement data or they are biased the higher

the uncertainty of the model results. Modelers could provide a more detailed answer to the question.

Line 62/63: information on emissions from Western Europe: looking at Fig.2 the given statement is supported, however in case of northerly/easterly/southerly conditions, the eastern stations can also provide emission information from other parts of Europe. In such conditions it also serves as a regional "background" to western/central Europe in model studies and inversions.

Yes, the Referee is absolutely right. However, in this general introduction, we could not go into the analysis of atypical meteorological situations because such a detour would have destroyed the structure of the section.

Line 68: delete "the" before "atmospheric". Consider adding at the end of sentence ", which are also reported to WDCGG".

Corrected. The sentence has been moved into Section 2.2 as Referee#1 requested.

2 Measurements and data

Line 78: high spatial representativeness: is it possible to give a quantitative measure or further description what is meant be "high" spatial representativeness?

According to the ICOS STILT model, the most important 50 % sensitivity area of the station (115 m) is 355 thousand $km^2$. This information is given in the caption of Fig. 2 of the revised manuscript.

Section 2.2. Monitoring system: Are any type of inlet filter used, to e.g. filter aerosol? If so, please provide technical details. If no filters are used, please comment on how fast/slow contamination by aerosol or particulate matter occurs in the lines or the KNF pumps.

The pumps are protected by Whatman GF/A glass microfiber filters. This information is inserted in the revised manuscript.

Line 157/158: please provide details on the typical ratio of bypass/sample flow

The sample flow rate was regulated by the Picarro analyzer itself, and it was not directly measured in the system presented. The important step was to have a stable overflow through the bypass to prevent the contamination of the sample air. The bypass flow was set to ~200 $cm^3$ $min^{-1}$. This value is given in the revised manuscript. In the measurement system rebuilt in 2023 for ICOS, the sample flow rate is also measured, and it is around 180 $cm^3$ $min^{-1}$.

Line 175-178: it could be useful to introduce, differentiate and define terms like e.g. "sampling uncertainty, which is a result of non-continuous sampling of the atmospheric variability within an hour" and "measurement, i.e. analytical, uncertainty". The description "high frequency sampling" can potentially be misleading, as not e.g. 10 Hz data logging is meant but the cycling through to each inlet height in sequence. Consider rephrasing the sentence.

The uncertainty in the hourly average concentrations due to the discontinuous sampling at the intakes is discussed in the referenced paper. The term "high-frequency sampling" may indeed be misinterpreted, therefore, the sentence has been rephrased.

Line 180: see also previous point, consider being more specific, e.g. "sampling time/switching frequency through inlet heights"

The sentence has been rephrased.

Line 187: low-uncertainty data: in the context of the section it seems clear that reference is made to the sampling uncertainty that arises out of the large atmospheric variability when e.g. the night time boundary layer breaks up. However, the way it is presented here could give the impression that the data from boundary layer transition periods is not used in models because of the above-mentioned sampling uncertainty. This however is not the main reason and thus consider rephrasing these sentences.

The misleading sentence has been removed.

Line 188: delete "when the demand appears". This statement is a home truth, but possibly worth reiterating.

The sentence has been removed.

Line 191: consider changing adjective "remarkable" to "great", quantifying also e.g. from x to xx m.

"Remarkable" has been changed to "great", and a range has been given in the revised manuscript.

Section 2.4. An extended section and assessment on the NOAA flask and in situ measurement comparison would be welcome.

An extended evaluation of the flask samples would go beyond the scope of the present study. The reasonable solution would be a joint work with NOAA covering flask sampling sites under different environmental conditions. At Hegyhátsál, the rigorous comparison of the flask and in situ measurements is hindered by the different locations of the intakes (flask: 96 m, in situ: 82 m and 115 m) and the temporal asynchrony between the flask samples and the in situ samples. For example, when the flask sample is taken at 96 m the in situ system may be sampling at 10 m.

Line 236: in the larger than 3 sigma cases, the presumed reasons could be further investigated and reasons confirmed, e.g. utilising the high-resolution continuous data during the flask sampling periods.

There are no high-resolution continuous data for the periods when the flasks are gradually filled up. The in situ system changes the intake sampled every two minutes, and only 30 s of data are available from each intake in each sampling period. In addition to the temporal asynchrony of the sampling, the different locations of the flask sampling intake and the in situ intakes cause problems. Only a statistical comparison of the two methods is possible. The

causes of >3σ deviations can only be identified with a reasonable probability in a few obvious cases.

Line 237: please specify the time period over which the mean deviation is calculated. Is the comparison robust throughout that time frame, or are there periods (years?)when the WMO compatibility goal is not achieved? Can a difference be seen for the different types of analysers and/or setups? Can a significant relationship be observed for larger flask-in situ differences when atmospheric variability is high?

The revised manuscript explicitly says that the comparison was made over the entire period (1994-2023). As there are temporal and spatial asynchronies between the flask and in situ measurements, only the long-term statistical analyses are defensible. Any other attempt may be challenged.

Line 241: elaborate or give examples what is meant by "technical issues"

The small reduction in the bias and scatter could be caused not only by the instrument change at Hegyhátsál but also by any possible technical improvement at NOAA. The staff at Hegyhátsál was also changed at that time, which might have affected the sample quality. Therefore the sentence has been deleted.

Line 243: how many flask samples contribute to that mean and standard deviation?

As the statement can be questioned, the sentence has been deleted (see above).

Section 2.5. Data selection. Line 249/250: consider adding information in the supplementary material to support the statement made on the lack of directional difference, e.g. a wind rose showing the residuals of detrended CO2 concentrations

A reference to a paper showing the sectorial distribution of the $CO_2$ concentrations at Hegyhátsál has been inserted into the revised manuscript to support the statement.

3 Results and discussion

Section 3.1. Line 264 and Line 311-312, Fig S2 and statement on nighttime boundary layer heights: please comment on the uncertainties in the BL height determinationin the ECMWF ERA5 data product? The dynamics of the nighttime BL is given as main reason for the high summer amplitudes of CO2 (Fig 5), however it seems the effect of the daytime BL (and the dilution effect) will have a lot larger impact on the large CO2 amplitude.

The calculated boundary layer heights depend on the algorithm used, and each algorithm has its internal uncertainty. In this study, ERA5 boundary layer height data were accepted as they are. The $CO_2$ measurements show that the trend of the high concentrations (upper percentiles) is higher than that of the low percentiles. The high concentrations form exclusively during the night, therefore, we can state that the summer daily concentration amplitudes are dominated by the nighttime processes. In the deep, well-mixed daytime summer boundary layer the $CO_2$ concentration is close to the continental background, or at least not much lower.

Line 316/317: can the statement on the "increasing respiration due to significant increase in temperature" be further supported. Is the observed nighttime temperature increase consistent with the suggested corresponding increase in respiration rates (from literature or e.g. nearby flux measurement data)? Has the vegetation cover(type and cover spatially) around the station changed in the time period, and if so, could that play a role in the observed trends?

For a more quantitative support, a boundary layer budget model should be developed and applied. At Hegyhátsál the surface-atmosphere $CO_2$ flux is measured by an eddy covariance system mounted at 82 m elevation on the tower. It has been operational since 1997. According to the study of Barcza et al. (2020), ecosystem respiration shows a significant positive trend, which may support our hypothesis. Unfortunately, the study referred to does not evaluate the trend with monthly resolution. The monitoring site is located in an agricultural region where the cultivars in the individual small plots change every year. However, the overall mixture of the cultivars in the region does not change. It should also be mentioned that the footprint of the concentration measurements and the flux measurements are quite different. Nevertheless, the revised manuscript refers to the study of Barcza et al. as it makes our hypothesis plausible.

Line 321/322: although reference to Haszpra et al., 2015 is made, please briefly describe salient points of the aircraft campaign, e.g. how was the top of the boundary layer sampled/derived (aerosol, T-profiles)? How many flights contribute to that mean, and what is meant by the "top of the tower underestimates the mean planetary boundary layer"? What could be the reason for the observed mean differences with the aircraft-based measurements? Please also be more specific as to how these results can be "informative for those using models with coarse vertical resolution" (Line 324).

The vertical concentration profile along the tower is mentioned in the manuscript only for completeness, devoting only a few sentences to the topic without conclusions. The profiles do not show any trend, and they correspond to the theoretical expectations. The spatial representativeness of the measurements at the top of the tower is the highest. Referring to the aircraft measurements we mention that even the measurements at the top of the tower are somewhat biased from the mean boundary layer concentrations. The discussion of the vertical profile of the concentration is not a major point of the present study, therefore, the details of the aircraft measurement campaign do not fit into the structure of the manuscript. The details of the campaign are given in the referenced paper.

Section 3.2. Line 368/9: please provide a citation/reference for this statement

A reference to Figure 8 has been inserted.

Line 374/375: please provide data and/or a citation/reference for this statement. Also comment on the role of regional sources in winter, e.g. from fossil fuel combustion for heating and energy?

January and February are the coldest months in the region, therefore, changes in heating and energy demand cannot be the cause of the decreasing atmospheric $CO_2$ concentration from January. This statement has been inserted into the revised manuscript.

Line 381/381: delete "but to the author's knowledge, the role of the changes in the dynamics of the atmosphere has not yet been studied in details."

Although we think this sentence is important suggesting research in this understudied area, we have deleted it following the Referee's request.

Line 397: as general statement okay, however please be more location speci fi c. Are additional phenology data from e.g. the Hungarian Met Service or else available that could be used for comparison and support here?

The onset of the growing season is plant-specific, therefore only a general statement can be made for an extended region like the footprint area of the tall-tower concentration measurements. Not forgetting about the contribution of the dynamics of the atmosphere, the date of the spring zero-crossing is related to the beginning of the growing season. However, this relationship is only qualitative on this spatial scale.

Line 397-400: provide guidance as which papers are relevant for the region/Hungary, and/or provide additional ones that show trend in Hungary

We are not aware of any Hungarian or regional studies. The papers referenced here either study the phenomenon on the Northern Hemisphere midlatitude scale or contain information relevant to the region.

Line 408: statement is made that trend is not significant, please provide details on trend and p value

The trend and its probability level are given in the revised manuscript.

Line 421-423: please comment on the uncertainty that is involved in the seasonal detrending, that could affect the trend analysis and thus possible detection of small trends

Although we cannot comment on the uncertainty of the algorithm applied by the CCGCRV software, we have added the trend and its probability level to the revised manuscript.

Line 425: decreasing winter peaks: please consider other reasons for decreasing winter peaks such as e.g. mild winters (less heating), events such as economic slowdown in the region and recently Covid-19 pandemic.

In the revised manuscript the milder winters, the presumably decreasing heating emissions are also mentioned. The economic slowdowns in the mid-2000s and recently, as well as the COVID-19 pandemic can have little effect on the 30-year trend.

Line 427: vague statement, consider rephrasing

The statement has been deleted.

Section 3.3. Line 449: It is unclear what is meant by "emissions vary in much narrower ranges". The previous sentence contains information on atmospheric concentration, here emissions are mentioned, which does not directly follow on from the data from Hegyhátsál.

To link the two sentences we have inserted the following explanation into the revised manuscript: globally 1 $\mu$mol mol$^{-1}$ increase corresponds to approximately 7,8 Pg $CO_2$ net input into the atmosphere.

Figure 12: please double-check the values of the growth rates for Hegyhátsál. Consider putting them in context with the global growth rate, as e.g. published in the WMO Greenhouse Gas Bulletin Nr.19 (Nov 2023) https://library.wmo.int/records/item/68532-no-19-15-november-2023?off set=1 . Global growth rates have been above 1 umol/mol/year since 1994, in the order of 1 to ~3.2 umol/mol/year.

The revised Fig. 12 also shows the temporal variation of the global growth rate calculated from the globally averaged marine surface monthly mean data (https://gml.noaa.gov/webdata/ccgg/trends/co2/co2_mm_gl.txt) for comparison. The growth rate data for Hegyhátsál in the figure have been checked and found correct.

Line 466-485: this section describes the observed pattern with ENSO, detailing lag times, however not sufficient detail is given what causes the effect on CO2 and how this information can be utilised and for whom it is relevant?

The purpose of this section is to show that the effect of a large-scale regional phenomenon in the tropical/southern subtropical Pacific Ocean can also be clearly detected in continental Europe despite the large distance and huge regional anthropogenic emissions. The mechanism of the teleconnection is not discussed here because it is rather complex and not fully understood yet. ENSO seems to cause pressure anomalies in the Northern Hemisphere also influencing the North Atlantic Oscillation, which affects the weather, in Europe and elsewhere. The regional weather conditions affect the biospheric $CO_2$ uptake and release, which is reflected in the regional growth rate of $CO_2$ concentrations. Beverly et al. (2024) recently published a paper listing the proposed ENSO-Europe teleconnection mechanisms.

Beverley, J. D., Collins, M., Lambert, F. H., and Chadwick, R.: Drivers of changes to the ENSO–Europe teleconnection under future warming, Geophysical Research Letters, 51, e2023GL107957, https://doi.org/https://doi.org/10.1029/2023GL107957, 2024.

4 Summary

Consider changing the summary to a conclusion, highlighting e.g. the value in such long-term observational data time series, their usage by others (modellers…) etc. This would strengthen the manuscript.

It is not easy to draw conclusions from an essentially descriptive review study. The main conclusion is that the slow tendencies presented in the paper could not be revealed in a shorter time series. Therefore, the long-term, uninterrupted operation of the monitoring sites is essential for a better understanding of the processes, and for revealing interactions and feedbacks. We have tried to reformulate the section accordingly.

Technical corrections

Abstract:

Line 9: be specific: "GAW ID code: HUN". Consider adding WIGOS station identifier 0-348-4-16307, here or later in Section 2.1

The station identification codes have been moved from the abstract to Section 2.1 where the WIGOS station identifier is also given in the revised manuscript.

1 Introduction

Line 19: consider changing "raised" to "suggested"

Corrected.

Line 22: add "that" before "it might"

Corrected.

Line 29: delete "However," starting the sentence with "One of the main (…)"

We feel that "however" is needed to emphasize the contradiction between the original concept (as far from the biosphere as possible) and the need to understand the role of the biosphere in the global carbon budget.

Line 31: insert "background" after "global"

Corrected.

Line 45: exchange "several" with "many"

Corrected.

Line 63: exchange "development" with "expansion"

Corrected.

2 Measurements and data

Line 83: delete square brackets, and add "which includes e.g." "after 6% other". Delete comma and "etc" after "settlements"

Corrected.

Line 84: "lessivated" without capitalised L

Corrected.

Line 85: delete comma after "Alfisol"

Corrected.

Line 91: unclear what type of road "2x1-line" describes, how many lanes?

Corrected. (2-lane, one lane in each direction.)

Line 169: consider changing "multi-elevation monitoring site" to "multi inlet height monitoring site"

Corrected.

Line 183/184: change to "Today models try to avoid these transition periods (…)"

Corrected.

Line 184: change to "CO2 concentration tends to be low."

Corrected.

Line 198: exchange "ventilated" with "flushed"

Corrected.

Line 200/201: change "Intake" to "intake", not capitalising the word

The sentence has been rephrased following the suggestion of the other Referee.

Line 219: delete one full stop at the end of sentence

Corrected.

3 Results and discussion

Line 323: in second square brackets delete comma before "2"

Corrected.

Line 354: delete "overlying"

Corrected.

Line 401: first use of "NDVI", please spell out

Corrected.

Line 436: consider changing "permanent" to "steady"

Corrected.

Line 446: change "bend" to "band"

Corrected.

Line 468: change "Barring Head" to "Baring Head"

Corrected.

Line 471: change "Plateau Rose" to "Plateau Rosa"

Corrected.

Line 492: delete "yet"

The section has been completely rewritten.

---

## Author Response (AR2)

Dear Editor,

Thank you very much for reviewing the revised version of our manuscript.

- Concerning the use of the monitoring data, following your recommendation, we refer to the paper of Friedlingstein et al. (2023) as an example instead of the Global Carbon Project website. This paper describes the methodology for the calculation of the atmospheric $CO_2$ budget from the concentration measurements (and other data).

- You are right, the low average bias does not necessarily mean that the measurements continuously met the WMO compatibility goal, therefore, so this statement has been deleted. I think details of the flask – in situ comparison would not fit into the main text, therefore, the new figures have been added to the Supplement. One of the figures shows all available data, while the other shows the average bias and the scatter of the data aggregated into 5-year bins. In addition to the statistical analysis, not much can be said about the data. In the first figure (Fig. S2 upper panel), it can be seen that the scatter of the data and the number of extreme outliers were higher when the Li-Cor 6251 analyzer was used until 2007. Both the change to the LI-7000 analyzer and a more trustworthy technician might contribute to the lower scatter and a lower number of extreme outliers after 2007 but we do not know anything about the technical changes at NOAA. The reasons for the recent increase in the scatter and the appearance of a few outliers are not clear yet. Most of the extreme outliers do not correlate with the variability of the concentration in the time window used for the comparison. This suggests that the extreme outliers are likely caused by sampling (e.g. contamination, improper flushing, misdating of the sample, etc.) or analytical errors, and the omission of these data is justified.
The second figure (Fig. S2 lower panel) aggregates the bias and the scatter into 5-year bins. Not considering the first period, the 5-year average biases are always within the WMO extended compatibility goal of 0.2 µmol mol$^{-1}$. This indicates at least that the measurements at Hegyhátsál are on the NOAA scale and that the quality of the measurements is fairly consistent throughout the monitoring period.

- The reference to Barcza et al. (2020) has been rephrased and extended.